# Patterns of Biodynamic Milk Fatty Acid Composition Explained by A Climate-Geographical Approach

**DOI:** 10.3390/ani9030111

**Published:** 2019-03-22

**Authors:** Ton Baars, Jenifer Wohlers, Carsten Rohrer, Stefan Lorkowski, Gerhard Jahreis

**Affiliations:** 1Research Institute of Organic Agriculture (FiBL), Ackerstrasse 113, 5070 Frick, Switzerland; 2KWALIS Qualitätsforschung Fulda GmbH, Fuldaer Str. 21, 36160 Dipperz, Germany; jeniferwohlers@web.de; 3Competence Cluster for Nutrition and Cardiovascular Health (nutriCARD) Halle-Jena-Leipzig, 07743 Jena, Germany; carsten.rohrer@uni-jena.de (C.R.); stefan.lorkowski@uni-jena.de (S.L.); gerhard.jahreis@uni-jena.de (G.J.); 4Department of Nutritional Biochemistry and Physiology, Institute of Nutritional Sciences, Friedrich Schiller University Jena, Dornburger Straße 25, 07743 Jena, Germany; 5Department of Nutritional Physiology, Institute of Nutritional Sciences, Friedrich Schiller University Jena, Dornburger Straße 24, 07743 Jena, Germany

**Keywords:** biodynamic milk, fatty acids, farm milk, shop milk, region of production, Europe, roughage intake

## Abstract

**Simple Summary:**

Biodynamic (BD) farming is one of the earliest organic labels. BD dairy farmers act according strict regulations on feed, fodder and manure cycling and their farms are known as low-input systems. The milk fatty acid (FA) composition of European BD farm milk was investigated in relation to its region of production. Farms were located in different climate zones. The FA composition was different between summer and winter, and increased levels of unsaturated FA were found in summer milk. Differences in milk FA between the three main regions (Atlantic, Central and Pre-Alpine) were caused by differences in rainfall, farm elevation and the length of the grazing season. The results are along the same line of knowledge how fresh grass, conserved fodder, maize silage and concentrates affect the milk FA profile. An important health marker for milk fat is the omega-6 to omega-3 (n6/n3) ratio, which should preferably be low. Milk fat based on pure grazing had a n6/n3 ratio of 1.00. Average BD shop milk had a lower n6/n3 ratio (1.37) compared to conventional shop milk (1.89). Based on the n6/n3 ratio, a BD dairy cow had a high intake (>82%) of fresh grass and conserved roughage (hay and grass silage).

**Abstract:**

Background: Biodynamic dairy production is based on a land-related animal production without the additional input of N-fertilizers. The concentrate level per cow is low. This affects the yield level of animals and product quality outcomes. Methods: We examined the milk fatty acid (FA) composition of European biodynamic farms in relation to the ecological region of production and the farm’s climate conditions. Climate data were derived from existing maps describing ecological vegetation zones within Europe. Additionally, biodynamic shop milk was compared to conventional shop milk, based on a regional comparison. Results: The largest differences in the FA composition were between biodynamic summer and winter milk. We found increased proportions of conjugated linoleic acid (CLA), alpha-linolenic acid (ALA-n3), monounsaturated FA (MUFA), and polyunsaturated FA (PUFA) in the summer milk. A principal component analysis expressed the structure that was present in the biodynamic farm milk samples, based on clusters of a single FA within four components. The components could be correlated with the season of production, the amount of precipitation, the elevation of the farm above sea level, and the length of the grazing season. Biodynamic shop milk in the summer had a lower n6/n3 PUFA ratio compared to the conventional shop milk in all regions of production. Mean values were 1.37 and 1.89, respectively. Conclusions: The differentiation of biodynamic milk FA composition is consistent with the existing knowledge about the effects of fresh grass, fodder, and ratio composition on the milk’s FA composition. Based on the n6/n3 PUFA ratio, the average biodynamic dairy cow had a high intake (>82%) of fresh grass and conserved roughage (hay and grass silage), especially in the summer.

## 1. Introduction

Biodynamic farming was one of the earliest movements in organic agriculture [1]. Biodynamic agriculture developed its own additional worldwide standards in addition to the regulations for organic agriculture. Biodynamics maintains the ecological principle of the farm as an organism. This limits the farmer’s ability to intensify the yield per hectare, as well as per animal, and the cycling of fodder and manure should be fulfilled. With regard to animal production, animals are kept according to their specific integrity. This means that ruminating animals are largely fed with roughage, not concentrates, whereas hay should always be part of the diet [2,3]. In comparison to conventional farming systems, the milk production per animal and per hectare is lower [4,5]. Biodynamic dairy production can be found in different areas of Europe; however, it is unknown whether the region of production impacts the farmer’s choices in a similar way to conventional farming, and how this affects the milk’s fatty acid (FA) composition. Farmers adapt their systems, their grazing policy, the length of the grazing period, and their choices for additional fodder according to the local climate conditions.

The level of n3 FA was found to differ between organic milk and conventional milk. The separation was more pronounced when the region of production was taken into account and stable carbon isotopes were integrated [6]. In the comparison of the FA composition of conventional and biodynamic milk in southern Germany, the differentiation between the farm origin (conventional versus biodynamic) depended on the level of intensification (low-input versus high-input). The authors showed that there was a complete separation between low-input biodynamic and high-input conventional milk FA composition. However, the milk FA patterns from high-input biodynamic production and low-input conventional production were overlapping [5]. The different intakes of fresh green fodder between conventional and organic cows played an important role, affecting the improved FA profile in organic milk fat (more: n3 FA, conjugated linolenic acids, vaccenic acid; less: palmitic acid) [7]. In grass-based milk production in New Zealand, differences in the milk’s FA composition depended on the intake of grass rather than on the label of production [8]. The importance of fresh grass intake was confirmed in recent studies in the US. Grass-based farming systems had more favorable FA profiles in terms of the conjugated linoleic acid (CLA) and n6/n3 polyunsaturated FA (PUFA) ratio compared to organic systems, due to the principle of roughage-based farming without additional concentrates [9]. If farmers produced milk from fresh grass and grass-based winter forage, a mean yearly n6/n3 PUFA ratio of 0.99 was reached. From a health perspective, westernized diets are overloaded with n6 FA and contain suboptimal levels of n3 PUFA. The conventional milk produced in feedlots can reach an n6/n3 FA ratio of 5.8, according to Reference [9].

Farming systems are very different across Europe. Soil type, slope, and the local climate impact the structure of the farm in terms of crops, grazing policy or length of the grazing season [10]. An increased distance to the Atlantic Ocean results in greater temperature extremes (day/night, summer/winter) and a shorter growing season [11]. Typical grassland areas in Europe are found in the coastal areas, because of the mild climate and higher precipitation, as well as in the mountainous and alpine regions [12]. The impact of the abiotic circumstances, climate conditions, and crop growth is also reflected in the natural vegetation zones across Europe [13,14], and it is assumed that both the natural vegetation and the farming system are affected by this. The aim of this study was to analyze the differences in the milk FA composition of biodynamic farms in Central and Northern Europe and to correlate the FA patterns with the geological and ecological background data of the farm location. A second aim was to compare biodynamic shop milk with conventional shop milk. The n6/n3 PUFA ratio of milk was taken as the main indicator as described in Reference [9] to estimate the impact of the label of production (biodynamic versus conventional).

## 2. Materials and Methods

Bulk tank milk samples: From different research projects conducted from 2008 to 2015, cow’s bulk tanker milk (N = 163) was sampled in biodynamic farms (N = 41). Samples were immediately cooled at 4 °C, transported, and frozen within 48 hours. All samples were analyzed at the Institute of Nutritional Sciences of the Friedrich Schiller University in Jena, following the same protocol over a period of 5 years. Bulk tank milk samples from published data [5,15,16] and unpublished data [17] were used for this purpose.

Shop milk samples: In September 2010, shop milk from both biodynamic (N = 10) and conventional farms (N = 10) was sampled in Sweden, Denmark, Germany, The Netherlands, and Switzerland. Samples were cooled at 4 °C and were frozen within 48 hours. All samples were analyzed at the Institute of Nutritional Sciences of the Friedrich Schiller University in Jena. Based on the creamery that delivered the shop milk, the location of processing in terms of elevation, latitude, and altitude was connected with the FA outcome. 

Lipid extraction and preparation of fatty acid methyl esters (FAME) and analysis by gas chromatography (GC): Frozen milk samples were thawed at room temperature and freeze-dried (fd60-1; Pharma & Food, Dresden, Germany). Milk powder was used for Soxhlet extraction using a Soxtherm2000 S306 A (Sartorius, Göttingen, Germany). FAME was prepared using NaOCH_3_. GC procedures were used to resolve all FA as described in Reference [18]. Briefly, for the separation of C4–C22, a fused-silica capillary column of medium polarity was used (GC-17 V3, Shimadzu, Kyoto, Japan; DB-225 MS: 60 m × 0.25 mm i.d., with 0.25 μm film thickness, Agilent Technologies, Palo Alto, CA, USA). Oven temperature was initially maintained for 2 min at 70 °C, then it was increased by 10 °C min^−1^ to 180 °C, and then further increased by 2 °C min^−1^ to 220 °C, and held for 5 min. In the final step, the temperature was increased by 2 °C min^−1^ to 230 °C and held for 27 min. The cis and trans isomers of C18:1 were separated using a fused-silica capillary column of high polarity (GC-2010 plus, Shimadzu; CP-select for FAME; 200 m × 0.25 mm i.d., with 0.25 μm film thickness, Varian, Houten, Netherlands). These isomers were separated under isothermal conditions at 176 °C. For the GC analysis, 1 μL of 2% FAME in n-hexane was injected with a split ratio of 1:100. For both procedures, the injector and detector temperatures were maintained at 260 and 270 °C, respectively. The carrier gas was hydrogen.

Farm ecology and geography: All the farms under study were located in Europe. Abiotic farm environmental and climate data were derived from the ecological vegetation zones in Europe described in References [13,19]. The farm location was pinpointed in Google maps, and longitude, altitude, and elevation above sea level were notified. The map of the environmental stratification of Europe (EnS), which included 84 environmental strata [12], was connected with the farm location. For each of the 84 strata, monthly values were present for five climate variables over the period from 1900 to 2000 [12,13]. For the purpose of this study, mean values of the following three variables over the period from 1960 to 1990 were used: -Monthly precipitation and year sum (mm);-Monthly average temperature and year mean (°C);-The monthly number of sun hours and year sum (N).

In addition, agronomic indicators were present for:-The length of the growing season (N number of days warmer than 5 °C), and-The annual temperature (°C) sums expressed as growing degree days with a 0 °C base (GDD_0_), as described in Reference [12].

Statistics: Single farm milk samples (N = 163) were obtained from 41 different farms. The number of samples per farm ranged between 1 and 27. Based on the date of sampling, milk was divided into summer (April to October) or winter (November to March) samples. This separation was further adjusted based on additional information of the cow’s rations, if present. All statistical calculation was done using SPSS version 20 (IBM SPSS Statistics for Mac, IBM Corporation, Armonk, NY, USA). The main data exploration was based on a principal component analysis following a stepwise linear regression. Based on the Scree plot structure, and after eliminating single FA with an expression lower than 0.3, four components were derived after Varimax rotation. The impact of the region of production was shown in a bi-plot. The values of the three main regions of production were rescaled based on the explained variance within each component. The linear data from the environmental background were correlated with the expressed factor scores of the four components based on a Pearson correlation and two-tailed significance. There were no fodder data present for each single milk sample. Based on the existing information between the correlation of fresh grass intake and the n6/n3 PUFA ratio in milk as in Reference [5], the amount of fresh grass intake was estimated for each of the milk samples. 

Shop milk samples from biodynamic and conventional origins were compared based on a paired *T*-test. Results were considered significant if *p* ≤ 0.05. 

## 3. Results

### 3.1. Farm Geodata Per Region

Farms were located in the northwestern part of Europe. The farms belonged to 14 of the 84 stratification zones within the European environmental zones (Table 1). For statistical evaluation, the 14 zones were reduced into three main areas of production: Atlantic, Central, and Pre-Alpine. Owing to the low number of Alpine farms (N = 2), these samples were included in the Pre-Alpine region. The farm elevation, precipitation, and longitude finally defined in which of the three main areas (Atlantic, Central or Pre-Alpine), the farms were positioned.

There were some characteristic climatic and geographical differences between the three main regions of production (Table 2). Besides the existing difference in elevation, latitude, and longitude, farms in the Pre-Alpine region had the highest precipitation, and together with the Central region, a lower temperature and more hours of sunshine. Typically, the Atlantic region had a higher mean temperature, the long growing season but with fewer hours of sunshine. The Central region had the lowest precipitation and the shortest growing season. Differences per month were not shown, but in the Atlantic region, there is a pretty constant amount of precipitation over the course of the year. The Central region showed the lowest monthly precipitation, especially outside the growing season (November to March). The Pre-Alpine and Atlantic regions stay coolest in summer, whereas the Central region showed the largest increase from winter to summer with the highest temperatures from May to August. The Atlantic region is mildest outside the growing season (November till March) but is at a similar level in the summer as the Pre-Alpine area. Cold impacts in the winter are mainly in the Central and Pre-Alpine regions. Most hours of sunshine are found in the Pre-Alpine region. Values per month are highest throughout the year, except for the months from April to June. The Atlantic region is lowest and showed the same line of development as the Central area.

### 3.2. Farm Milk Fatty Acid Composition

In Table 3, the FAs are listed based on the 85 summer and 78 winter samples. 

There were typical differences in the milk FA composition between the samples taken within or outside the grazing season (summer or winter milk). Summer milk showed higher levels of PUFA and monounsaturated FA (MUFA), particularly caused by the elevated levels of all C18:1t and C18:1c FA, CLAc9t11, CLAt11c13, and C18:3c9c12c15. Within the short-chain FA (SCFA), the even chain numbers (C4:0–C8:0) were higher in the summer milk, whereas the odd-chain FA (C7:0–C11:0) were higher in the winter milk. 

After a principal component analysis plus Varimax rotation, the structure of the data was split into four components, explaining 62.6% of the total variance (Table 4). The sum FA were correlated with the factor scores of the four components (Table 5), as well as the geological, climate, and ecological data (Table 6). Given the interest in the impact of the region on the FA outcomes, the results of the 2nd against the 4th component were graphically presented in a bi-plot (Figure 1). 

Most of the explained variance was in the first component (22.6%). The differentiation upon this axis showed a very high correlation with the season of production (Table 6). To a smaller extent, the information on this axis was controlled by the region of production: both the Pre-Alpine and Atlantic versus the Central origins. C18:0 and C18:1 FA were increased in the summer milk, whereas the medium-chain FA (C12:0–C16:0) were increased in the winter milk. For the environmental factors, there was a correlation with the amount of precipitation and the elevation level of the farm. The correlation with the sum FA was based on the differentiation between the MUFA, C18:1t, and CLAs to the milk from the summer and the short- and medium-chain FA in the winter milk.

The second component (16.0% of the variance explained) was based on the load of the medium-chain saturated FA (C7:0–C12:0). This was correlated with the contrast between the milk from the Atlantic and the Pre-Alpine regions. Milk in the Pre-Alpine region was produced at a higher elevation, in areas with higher precipitation and more hours of sunshine than in the Atlantic regions. For the sum FA, the level of saturated FA, as well as the short- and medium-chain FA, were increased in the milk from the Atlantic farms, whereas the MUFA and n6 PUFA increased in the milk from the Pre-Alpine farms.

The third component (14.9% of the variance explained) represented the odd- and branched- chain FA, as well as several n3-FA. The contrast was based on milk from the Atlantic regions compared to milk from the Central region in the Northern part of the research area. Milk from the Atlantic farms was produced in milder areas, with a longer growing season and a higher average yearly temperature, but with fewer hours of sunshine than in the Central region. Not shown was the correlation with the single month, but there were significant correlations especially with the higher temperatures in autumn and winter, indicating mild winters (Pearson correlation = 0.35) and a longer period of grass growth. For the sum FA, there was a strong load of n6 FA, as well as a high n6/n3 ratio, for milk from the Central area, whereas the n3 was higher in the Atlantic farms.

The fourth component (9.1% of the variance explained) was built on the contrast between the two n6-PUFA plus C16:1t9, and CLAc9t11, CLAc11t13, and C18:1t11. There was a strong positive correlation with the Central region (n6 PUFA) and a negative correlation with the Pre-Alpine region (C16:1t9). The amount of precipitation, the elevation level, and the amount of sunshine were higher in the Pre-Alpine region compared to the Central region. The correlation with the sum FA revealed an increased PUFA, CLA, C18:1t, and n3 in milk from the Pre-Alpine region and a higher amount of short-chain FA plus a higher n6/n3 PUFA ratio in milk from the Central region. Not shown was the significant correlation with a higher average monthly temperature from April to September in the Central region.

### 3.3. Bi-plot Expressing The Impact of The Region of Production

The impact of the region of production on biodynamic farm milk FA composition, and its underlying climate and geographic factors, is summarized in Figure 1. The comparison of the second and fourth components of Table 6 is expressed. The second component differentiates between the Atlantic and Pre-Alpine regions, whereas the fourth component differentiates between the Central and Pre-Alpine regions. There was a significant correlation with latitude and elevation, indicating that both the Atlantic and Central regions were predominantly in the Northern part of the research area, as compared to the Pre-Alpine farms, which were located at higher elevations in the mountains. For the climate data, the mean temperature during the growing season (May–October) was higher for the Central region, whereas the higher yearly average rainfall and hours of sunshine were correlated with the Pre-Alpine region. Climate and geography both affected the FA characteristics of the biodynamic milk in each region. Across the horizontal axis (2nd component), higher levels of short-chain (SC) and medium-chain saturated FA (MC SFA) and lower levels of long-chain saturated FA (LC SFA) were key characteristics of the Atlantic milk. Within MUFA, Atlantic milk was based on MC cis-MUFA, whereas LC cis-MUFA was limited. Across the vertical axis (4th component), the differentiation between Pre-Alpine and Central milk was based on higher levels of CLAs and its MUFA-precursors (C16:1t9 and C18:1t11), odd LC SFA, and several n3 PUFA in the Pre-Alpine milk. Meanwhile, the Central milk had higher levels of even LC SFA, several other t-MUFA (C18:1t9, C18:1t10), and several n6 FA. Odd- and branched-chain FA (OBFA) were largely uncorrelated with either region.

### 3.4. Comparison of the Shop Milk of Biodynamic And Conventional Origin

Based on the location of the creamery that processed the milk, there were no differences in elevation between the two labels (Table 7). In shop milk, the n3 FA (ALA, C20:5n3, and C22:5n3) and CLAs (CLAc9t11 and CLAt11c13) and its precursors (C18:1t11 and C16:1t9) were higher in the biodynamic milk. The largest difference between the two groups was found for C18:1c15, which was higher in the biodynamic milk. In contrast, in conventional milk samples, the proportions of saturated odd, short-chained FA (C7:0 and C11:0), as well as cis medium-chain FA (C16:1c9 and C18:1c12), were higher.

### 3.5. Comparison of Shop Milk And Farm Milk in The Summer

In Figure 2, the relationship between the elevation of the creamery or the farm and the n6/n3 PUFA ratio in milk is shown. Biodynamic shop milk and biodynamic farm milk showed a very similar regression line, which indicated that the shop milk samples roughly reflected the results of the farm milk samples. In the comparison of the regression lines of the biodynamic shop milk and the conventional shop milk, the slope of the line looked similar, but the line was around 0.6 points higher in the conventional shop milk. 

### 3.6. Calculated Intake of Grass And Roughage

Based on an equation between the intake of grass-based products (x) and the n6/n3 PUFA ratio (y), which was derived from the data in Reference [5]: y = 0.000012x^3^ − 0.0025x^2^ + 0.1204x + 1.9206; on average the biodynamic milk was produced on the basis of calculated intake of grass products, that is, 86% grass products in the summer and 82% grass products in the winter. Higher roughage intakes could be calculated in the milk from higher altitudes.

## 4. Discussion

In this study, the FA composition of 163 biodynamic milk samples taken across Europe was analyzed. The patterns in the FA composition were correlated with the ecological and climate data of the farm environment. Based on a principal component analysis, 63% of the variance was explained by four components. The different FA patterns could be correlated with the season of production, precipitation level, the farm elevation, and the length of the growing season. In a small set of paired shop milk samples, a more favorable FA profile in relation to health was found in the biodynamic milk compared with the conventional milk. The ratio n6/n3 PUFA in both the biodynamic shop milk and biodynamic farm milk had a similar correlation with the elevation level of production, whilst the conventional shop milk was different. Based on the n6/n3 ratio, biodynamic milk was based on a >82% roughage and grass intake by the cows.

For the grass growth across Europe, productivity was closely related to the annual precipitation and to a lesser extent to the mean temperature and length of the growing season [20]. Climate, soil type, slope, and elevation affected the farmer’s choice of the type of pasture, regular grassland renewal, and crop rotation. The growing pattern of the grassland and the start of the growing season depended on the climatic conditions. Early spring growth or even winter growth is found in the southern regions in Europe, in lower elevated areas, and in areas affected by the Gulf Stream (mild, rainy). To keep cows grazing during the summer, continuous growth of grass is necessary and is connected with a high amount of precipitation. Typical European grassland areas are found on the British continent, the coastal areas in Denmark, Northern-Germany, and The Netherlands, Normandy (France), Galicia (Spain), and the Alpine regions [20]. The continental climate is the typical area for mixed farming, where besides grass-legumes mixtures, cows can be fed with annual fodder crops like maize silage or whole plant silage. A high daily summer temperature forces farmers to reduce the cow’s grazing time. The growing conditions for fresh grass are poor due to the low precipitation and high evaporation and temperatures. Moreover, dairy cows reach the upper-temperature limit of their comfort zone, and therefore, are kept indoors [21]. When kept indoors, cows will often be fed with conserved forages and concentrates rather than fresh grass. In combination with a short grazing season, especially in more Nordic European regions, cows have a long winter period inside and rely on conserved forages and concentrates.

The highest precipitation and most sun hours were in the Pre-Alpine area in combination with the lowest average daily temperature. The climate of the Central region, together with the Pre-Alpine climate, led to the shortest growing season. In the Central region, the grass growth can be limited in the middle of the season, especially when the summer precipitation is low and temperatures are high. In the Pre-Alpine region in spring, there is a high speed of grass growth, after a relatively late start of the growing season. The time of this burst of grass growth takes place in relation to the elevation of the land. Typical also for the Pre-Alpine region is the high precipitation during the growing season. Comparable to the Atlantic area with its high precipitation, the choice of annual fodder plants is undesirable in the Pre-Alpine regions, also due to the slope of the fields. Particularly, in the Pre-Alpine area, farmers depend on permanent pastures, often rich in plant species [22].

Several authors [5,8,9] discussed the question of whether an organic label of production had more impact on the FA composition and specific FA markers (n3, n6/n3, CLA, and its precursors) than the overall grazing and feeding policy at the farm. A complete separation of farming systems was only possible if conventional farms had a high-input strategy and biodynamic farms a low-input strategy [5]. However, at grass-based farms in New Zealand, conventional farms produced a more favorable FA composition in terms of CLAc9t11 and its precursor C18:1t11 than organic farms [8], even though organic cows had more access to grass than conventional cows. In fresh grass, there is a linear relationship between the ALA concentration and the protein concentration [23]. Therefore, the explanation for the unexpected difference in New Zealand milk CLA levels might be due to the higher speed of grass growth after the grass received nitrogen fertilizer at the conventional farms [8]. Not only the access to fresh grass but also the constant creation of new plant cells affect the final level of CLAc9t11 and C18:1t11 in milk fat. In Reference [16], a high content of phytanic acid in the milk fat was a better prediction of the fresh grass intake. The concentration of phytanic acid was under the direct influence of the chlorophyll content of the cow’s feed, and a further differentiation could be made between fresh grass and grass silage on the one hand and dried hay on the other hand. A strong reciprocal correlation was found between the sum amount of concentrates plus maize silage in the cow’s diet and the phytanic acid level [16]. 

In this study, after the principal component analysis we have found four components to differentiate the 163 biodynamic milk samples. Other authors have expressed the total explained variance across only the first two, main components [5,8,24,25]. In Reference [25], the milk from French farms in a mountainous area in summer and winter was investigated, whereas, in Reference [8], the impact of the season and label (organic versus conventional) on the milk FA composition at New Zealand farms was compared. In Reference [25], the main separation was based on the seasonal impact, which was similar in our study. Grass feeding (summer) versus feeding of conserved forages (winter) increased the levels of C18:1c9 and C18:1t11, in general, all trans-isomers of C18:1, CLAc9t11, and C18:0. In our four-dimensional solution, the effect of a higher fresh grass intake was reflected in the outcomes of the components one and four covering the differentiation of summer and winter milk, respectively, and milk from the Pre-Alpine and Central regions. The amount of precipitation was an important abiotic factor differentiating these results, and especially in component four (Pre-Alpine versus Central), where the correlation was high. This implies that when water becomes restrictive for grass growth, like in the Central region, biodynamic farmers start feeding other forages and concentrates for maintaining their milk yield. In Reference [25], a correlation between a higher altitude in the mountains and the presence of permanent pastures was found. For our milk samples, the elevation was positively correlated with component one and negatively with components two and four, and the typical ‘Pre-Alpine effect’ was reflected in components two and four. If cows were only consuming semi-permanent forages and low amounts of concentrate [25], they would produce milk with the highest concentration of CLAc9t11 and ALA. This was reflected in component four.

According to Reference [25], the access to permanent pastures was correlated with the group of iso- and ante-iso FA from C15:0 and C17:0, as well as C18:3c9c12c15 (ALA) and C20:5n-3. This differentiation was in concordance with component three, differentiating the milk from the Atlantic and Central region. According to Reference [25], the branched-chain FA was not affected by the season, which was confirmed by our data. Based on these outcomes, permanent pastures, as well as access to grassland and grazing, is limited in Northern and Continental Europe. In Reference [26], a correlation between the increased amount of C14iso and the decrease of concentrates in the diet could be calculated. C14iso is part of the third component separating milk from the Atlantic and Central regions, suggesting higher concentrate levels at the Central farms to compensate for a lower forage intake.

In Reference [25], the intake of hay was solely correlated with C15:0. C15:0 is part of the third component, suggesting a higher intake of hay at the Atlantic farms compared to the Central farms. They showed that feeding of grass silage and concentrates was correlated with C12:0 and C14:0, C14:1c9 and C16:0, respectively. These results were found in component one, where they built the negative direction of the outcome, in accordance with the winter fodder situation. C16:0 was found to be a strong separator between the summer-grazing and winter-feeding period, which was reflected in the strong negative correlation of C16:0 in component one (−0.921). In Reference [25], maize silage was solely correlated with C18:1c12. In our analysis, C18:1c12 was found in relation to components one, three, and four. In components 3 three and four, C18:1c12 was found in connection with milk from the Central area. Reduced precipitation is one of the motivations for farmers to grow fodder maize. Farms relied more on a crop rotation of fodder plants, like whole plant silage, maize silage, and grass-legume silage as winter fodder, but also as supplementation for conserved fodders during summer. Biodynamic high input farms in the German Allgäu region had a reduced grass intake during summer in contrast to the biodynamic low-input farms [5]. Cows on biodynamic high-input farms produced more milk based on the supplemental feed from concentrates, grass silage, and maize silage compared to the biodynamic low-input farms. In addition, low-input farms were more dependent on roughage [15], and only used a limited amount of concentrates, as well as some hay [5].

Our results showed that biodynamic farm milk had a mean ratio n6/n3 PUFA in summer of 1.45 and 1.64 in the winter. When farmers have low access to grass and grazing, and when they depend on high yielding cows, they adapt the cow’s diet and apply concentrates, maize, and other conserved fodder rather than grass only. This choice impacts the n6/n3 ratio, which increased rapidly [16,23]. In a smaller sample set of milk samples taken in the German Allgäu region, all the conventional farms— low-input farms and high-input farms—fed higher amounts of concentrates per cow, and biodynamic farms were more dependent on roughage [5]. The roughage-based character of biodynamic milk production in other areas of Europe was confirmed in the present study. Based on the average n6/n3 ratio, the FA quality of biodynamic summer milk (1.45) was quite near to the FA quality of the US systems based on pure grazing (0.99), although there was quite some difference between the lowest and the highest values (Table 3).

## 5. Conclusions

Milk fat from biodynamic dairy cows showed a differentiation which was explained by the season and the region of production, as well as the local climate at the farm, especially the precipitation level and farm elevation level. The strongest separation in biodynamic milk FA composition was between summer and winter milk. The amount of precipitation is a consistent abiotic factor that affects the amount of fresh grass growth at the farm and the grass intake of cows. The differentiation of biodynamic milk FA composition is in accordance with the existing knowledge on how grass and grazing in summer, the choice of crops, as well as the composition of the roughage diet in winter affect the milk FA composition. Differences between the FA composition of biodynamic and conventional shop milk, based on the relationship of predicted roughage intake and the n6/n3-ratio in milk fat, point to the fact that biodynamic cows had, on average, a high intake of fresh grass, hay, or grass silage covering >82% of the cow’s diet.

## Figures and Tables

**Figure 1 animals-09-00111-f001:**
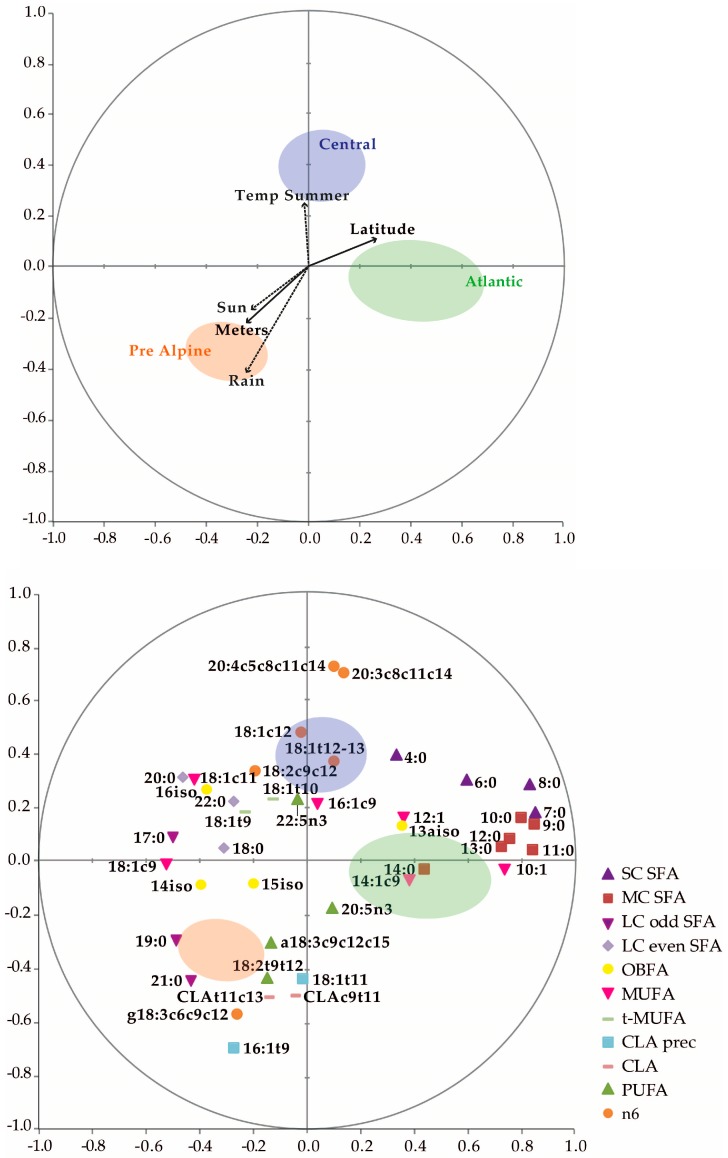
(**A**) (above) and (**B**) (below). Bi-plots of the second and fourth components after the principal component analysis, explaining 16.0 and 9.1% of the total variance, respectively. Fatty acid (FA) with a low load for both axes are not shown (<0.2). (**A**) Group means of the region of production ± 2SEM ellipses based on the rescaled axes plus the correlation with the significant geographic and climate data. (**B**) Single FA marked in different groups. Abbreviations: Meters = elevation above sea level; Latitude in °N; Sun = Mean sun hours per year; Rain = mean precipitation per year; Temp Summer = Mean temperature in the summer months; SC = short-chain; MC = medium-chain; LC = long-chain; SFA = saturated FA, OBFA = odd- and branched-chain FA; MUFA = monounsaturated FA; t-MUFA = trans-MUFA; CLA = Conjugated Linoleic Acid; CLA prec = precursor of CLA; PUFA = poly-unsaturated FA; n6 = omega 6 PUFA; n3 = omega 3 FA.

**Figure 2 animals-09-00111-f002:**
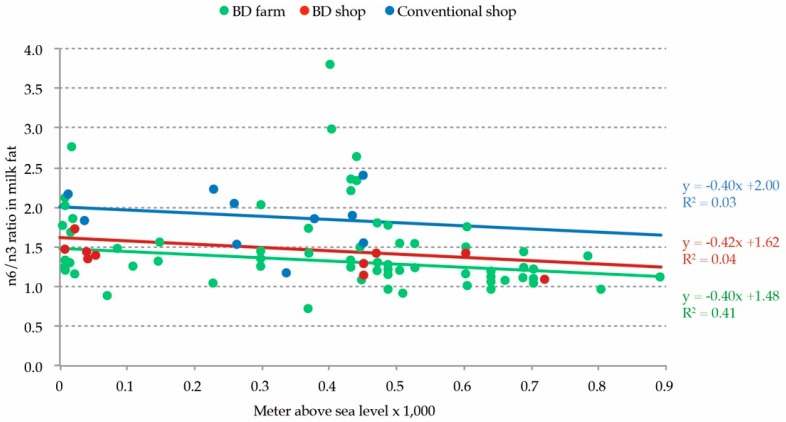
The relationship between the elevation of the creamery and the ratio of n6/n3 in the milk fat of the biodynamic (BD) summer shop milk (N = 10), conventional summer shop milk (N = 10), and biodynamic farm milk samples in the summer (N = 89).

**Table 1 animals-09-00111-t001:** Farm number, country, stratification area, and environmental zone according to Reference [12], followed by the clustering of the environmental zones into three main areas; farm position based on elevation, latitude, and longitude.

Farm	Country *	Stratification Area	Environmental Zone	Main Area	Elevation (m)	Latitude (°E)	Longitude (°N)
1	S	NEM5	Nemoral	Central	15	59.055	17.610
2	S	NEM5	Nemoral	Central	18	59.066	17.623
3	DK	ATN5	Atlantic-North	Atlantic	3	54.928	8.968
4	DK	ATN5	Atlantic-North	Atlantic	21	55.691	8.686
5	DK	CON9	Continental	Central	22	56.087	12.39
6	NL	ATC2	Atlantic-Central	Atlantic	-4	52.580	4.798
7	NL	ATC2	Atlantic-Central	Atlantic	16	52.011	6.414
8	PL	CON3	Continental	Central	146	53.677	16.492
9	D	ATC2	Atlantic-Central	Atlantic	149	50.698	6.712
10	D	ATC2	Atlantic-Central	Atlantic	196	49.968	6.857
11	D	ATN4	Atlantic-North	Atlantic	9	53.796	9.613
12	D	ATN4	Atlantic-North	Atlantic	167	51.949	9.917
13	D	ATN4	Atlantic-North	Atlantic	227	51.026	9.636
14	D	ATN4	Atlantic-North	Atlantic	299	51.394	9.003
15	D	ATN4	Atlantic-North	Atlantic	300	51.462	8.410
16	D	CON4	Continental	Central	369	49.200	9.935
17	D	CON4	Continental	Central	403	49.230	10.772
18	D	CON4	Continental	Central	442	49.300	10.772
19	D	CON4	Continental	Central	449	49.049	10.574
20	D	CON4	Continental	Central	504	48.683	10.130
21	D	CON5	Continental	Central	71	53.585	10.520
22	D	CON5	Continental	Central	86	52.292	14.024
23	D	CON5	Continental	Central	109	50.194	8.753
24	D	CON6	Continental	Pre-Alpine	432	48.025	7.585
25	D	CON6	Continental	Pre-Alpine	472	47.771	9.204
26	D	CON6	Continental	Pre-Alpine	486	48.112	12.723
27	D	CON6	Continental	Pre-Alpine	489	47.996	12.728
28	D	CON6	Continental	Pre-Alpine	527	48.200	10.383
29	D	CON6	Continental	Pre-Alpine	602	48.033	9.750
30	D	CON6	Continental	Pre-Alpine	606	47.799	9.144
31	D	CON6	Continental	Pre-Alpine	639	47.851	9.139
32	D	CON6	Continental	Pre-Alpine	689	47.756	9.740
33	D	CON6	Continental	Pre-Alpine	702	47.756	10.050
34	D	CON6	Continental	Pre-Alpine	804	47.751	10.268
35	L	CON6	Continental	Pre-Alpine	508	50.096	6.058
36	CH	ALS5	Alpine South	Pre-Alpine	1234	46.727	12.201
37	CH	ALS6	Alpine South	Pre-Alpine	1176	46.727	7.880
38	CH	CON2	Continental	Pre-Alpine	784	47.776	12.191
39	CH	CON2	Continental	Pre-Alpine	892	47.111	6.907
40	CH	CON2	Continental	Pre-Alpine	912	47.533	10.067
41	CH	CON6	Continental	Pre-Alpine	660	47.476	9.089

* CH = Switzerland, D = Germany, DK = Denmark, L = Luxemburg, NL = The Netherlands, PL = Poland, S = Sweden.

**Table 2 animals-09-00111-t002:** Geography and climatological values of the three regions of production; mean values per year and standard deviation (SD), based on the strata outlined in Reference [12].

Geography and Ecology	Region of Production			*p*-Value
	Atlantic (SD)	Central (SD)	Pre-Alpine (SD)	
Farms (N)	11	12	18	
Elevation (m above sea-level)	126 (121) ^a^	220 (195) ^a^	701 (232) ^b^	<0.001
Latitude (°E)	52.319 (1.791) ^b^	52.451 (3.875) ^b^	47.805 (0.713) ^a^	<0.001
Longitude (°N)	8.092 (1.649) ^a^	12.466 (3.169) ^b^	9.728 (1.908) ^a^	0.000
Growing degree days (°C)	3436 (224) ^b^	3221 (206) ^a^	3376 (210) ^a,b^	0.050
Length of growing season (day)	259 (22) ^c^	224 (11) ^a^	239 (11) ^b^	<0.001
Precipitation (mm)	748 (21) ^b^	641 (53) ^a^	1025 (59) ^c^	<0.001
Temperature (°C)	9.0 (0.6) ^b^	8.1 (0.8) ^a^	7.9 (1.0) ^a^	0.005
Sunshine (hours)	25.2 (2.5) ^a^	30.3 (2.1) ^b^	33.0 (3.5) ^c^	<0.001

^a,b,c^: Columns with different letters are statistically significant according the Tukey’s honestly significant differences HSD Test.

**Table 3 animals-09-00111-t003:** Single (above) and sum (below) fatty acid (FA) from the biodynamic summer (left) and winter (right) milk samples. Mean value, standard deviation (SD), minimum (Min), and maximum (Max), as well as the ratio from summer to winter (S-W)/Wx100; Samples: N = 163.

Single FA	Mean and SD, Minimum and Maximum						Ratio
	Summer (SD)	Min S	Max S	Winter (SD)	Min W	Max W	S to W
C4:0	3.79 (0.51)	2.24	4.96	3.18 (0.69)	1.21	4.15	19.0
C6:0	2.46 (0.30)	1.80	3.53	2.35 (0.39)	1.52	3.92	4.7
C7:0	0.04 (0.01)	0.01	0.08	0.04 (0.01)	0.01	0.09	−7.2
C8:0	1.33 (0.17)	0.91	1.79	1.29 (0.19)	0.83	1.74	3.0
C9:0	0.03 (0.01)	0.01	0.07	0.03 (0.01)	0.01	0.06	−10.5
C10:0	3.02 (0.51)	1.85	4.59	3.21 (0.42)	2.11	4.12	−5.8
C11:0	0.05 (0.02)	0.02	0.13	0.06 (0.02)	0.02	0.11	−11.2
C12:0	3.24 (0.57)	1.95	4.95	3.63 (0.50)	2.26	4.80	−10.7
C13:0	0.09 (0.02)	0.06	0.16	0.11 (0.02)	0.06	0.15	−11.2
C14:0	11.23 (0.90)	8.93	12.98	12.34 (1.06)	9.60	14.80	−9.0
C15:0	1.33 (0.14)	0.91	1.74	1.39 (0.18)	0.98	1.74	−4.6
C16:0	29.41 (2.51)	24.29	36.74	35.99 (4.27)	24.63	43.27	−18.3
C17:0	0.64 (0.07)	0.46	0.81	0.66 (0.08)	0.45	0.87	−2.1
C18:0	9.77 (1.15)	6.81	12.29	7.77 (1.89)	5.03	12.68	25.8
C19:0	0.07 (0.02)	0.04	0.13	0.08 (0.03)	0.01	0.16	−8.6
C20:0	0.15 (0.02)	0.11	0.25	0.15 (0.03)	0.09	0.24	−0.5
C21:0	0.02 (0.01)	0.01	0.05	0.02 (0.01)	0.00	0.05	−3.9
C22:0	0.07 (0.01)	0.04	0.11	0.07 (0.02)	0.02	0.11	1.0
C24:0	0.04 (0.01)	0.02	0.06	0.04 (0.01)	0.01	0.06	3.2
C13aiso	0.05 (0.02)	0.01	0.10	0.06 (0.03)	0.00	0.11	−26.7
C14iso	0.14 (0.03)	0.08	0.22	0.14 (0.03)	0.08	0.21	−6.1
C15iso	0.32 (0.06)	0.18	0.47	0.32 (0.07)	0.18	0.49	−1.2
C15aiso	0.58 (0.10)	0.37	0.81	0.56 (0.10)	0.37	0.79	5.1
C16iso	0.25 (0.03)	0.19	0.37	0.26 (0.04)	0.18	0.35	−1.5
C17iso	0.39 (0.05)	0.21	0.57	0.34 (0.05)	0.22	0.44	14.5
C17aiso	0.44 (0.06)	0.28	0.60	0.45 (0.06)	0.33	0.58	−0.8
C10:1	0.30 (0.05)	0.20	0.43	0.32 (0.06)	0.15	0.45	−6.0
C12:1	0.05 (0.03)	0.01	0.11	0.07 (0.03)	0.01	0.13	−25.9
C14:1c9	0.92 (0.15)	0.65	1.44	1.10 (0.24)	0.60	1.64	−16.2
C16:1t9	0.24 (0.12)	0.06	0.51	0.15 (0.08)	0.00	0.27	59.2
C16:1c9	1.36 (0.20)	0.97	1.83	1.64 (0.30)	1.10	2.48	−17.1
C18:1t4-8	0.15 (0.05)	0.01	0.31	0.10 (0.06)	0.02	0.29	59.4
C18:1t9	0.22 (0.04)	0.14	0.33	0.16 (0.06)	0.06	0.33	35.2
C18:1t10	0.19 (0.05)	0.09	0.38	0.13 (0.07)	0.03	0.33	43.0
C18:1t11	2.23 (0.79)	0.80	4.13	0.91 (0.34)	0.44	1.98	144.0
C18:1t12-13	0.36 (0.15)	0.08	0.91	0.26 (0.12)	0.02	0.61	40.9
C18:1t15	0.20 (0.06)	0.08	0.43	0.13 (0.08)	0.03	0.52	62.4
C18:1t16	0.38 (0.07)	0.19	0.54	0.22 (0.09)	0.03	0.47	73.7
C18:1c9	18.85 (1.82)	15.14	23.68	16.12 (3.43)	11.08	25.19	16.9
C18:1c11	0.52 (0.10)	0.32	0.80	0.45 (0.14)	0.22	0.86	14.1
C18:1c12	0.12 (0.07)	0.05	0.43	0.10 (0.06)	0.03	0.34	18.7
C18:1c13	0.10 (0.02)	0.03	0.17	0.06 (0.02)	0.00	0.14	57.9
C18:1c15	0.09 (0.04)	0.00	0.20	0.03 (0.03)	0.00	0.14	155.8
C20:1c11	0.05 (0.03)	0.01	0.15	0.04 (0.01)	0.02	0.07	42.1
C18:2t9t12	0.25 (0.16)	0.03	0.75	0.18 (0.12)	0.03	0.48	39.3
C18:2c9c12	1.12 (0.29)	0.65	2.15	1.09 (0.34)	0.51	2.40	3.2
CLAc9t11	1.24 (0.51)	0.40	2.80	0.59 (0.18)	0.32	1.22	111.3
CLAt11c13	0.08 (0.04)	0.01	0.22	0.04 (0.03)	0.01	0.15	120.2
C20:2c11c14	0.02 (0.01)	0.01	0.10	0.02 (0.01)	0.01	0.04	2.9
Gamma C18:3c6c9c12	0.10 (0.15)	0.01	0.74	0.14 (0.13)	0.01	0.56	−30.2
Alpha C18:3c9c12c15 (n3)	0.91 (0.27)	0.40	1.89	0.78 (0.23)	0.37	1.57	16.2
C20:3c8c11c14	0.07 (0.02)	0.04	0.12	0.06 (0.02)	0.03	0.12	3.5
C20:4c5c8c11c14	0.06 (0.02)	0.04	0.12	0.07 (0.02)	0.04	0.12	−5.7
C20:5 (n3)	0.09 (0.02)	0.04	0.13	0.08 (0.02)	0.03	0.12	8.8
C22:5 (n3)	0.10 (0.02)	0.04	0.13	0.10 (0.03)	0.04	0.15	−1.2
Sum FA							
Saturated FA	69.10 (2.69)	63.75	75.26	74.70 (4.13)	62.32	80.96	−7.5
Short-chain FA	7.62 (0.87)	5.18	9.99	6.86 (1.17)	3.72	9.84	11.0
Medium-chain FA	19.12 (1.96)	14.98	24.17	21.07 (2.09)	15.02	25.75	−9.3
Monounsaturated FA	26.42 (2.21)	21.74	30.94	22.16 (3.80)	17.20	32.71	19.3
Polyunsaturated FA	4.48 (0.85)	2.76	7.04	3.40 (0.58)	2.31	5.05	31.7
n3 FA	1.43 (0.33)	0.85	2.48	1.17 (0.28)	0.67	2.03	21.8
n6 FA	1.97 (0.42)	1.17	3.24	1.79 (0.57)	0.99	3.98	9.8
C18:1t	3.75 (0.89)	1.75	5.98	1.91 (0.64)	0.78	4.09	96.6
CLA	1.58 (0.57)	0.57	3.30	0.74 (0.24)	0.35	1.46	113.3
n6/n3	1.45 (0.51)	0.72	3.81	1.64 (0.80)	0.84	4.98	−11.6

**Table 4 animals-09-00111-t004:** Clustering of single fatty acid (FA) based on the four components after principal component analysis with Varimax rotation. Only loadings above 0.3 are shown. In the last line, the explained variance of each component is shown.

Single FA	Component			
	1	2	3	4
C4:0	0.532	0.326		0.388
C18:0	0.824	−0.309		
C18:1t4-8	0.776			
C18:1t9	0.803			
C18:1t10	0.705		−0.417	
C18:1t11	0.757			−0.444
C18:1t12-13	0.632		−0.311	0.364
C18:1t15	0.692			
C18:1t16	0.891			
C18:1c9	0.706	−0.521		
C18:1c11	0.502	−0.419		0.302
C18:1c13	0.775			
C18:1c15	0.662			
CLAc9t11	0.680			−0.505
CLAt11c13	0.550			−0.510
C12:1	−0.424	0.358		
C13aiso	−0.443	0.349		
C14:0	−0.706	0.437		
C14:1c9	−0.645	0.375		
C16:0	−0.921			
C16:1c9	−0.649			
C6:0		0.587		
C7:0		0.842		
C8:0		0.816		
C9:0		0.839		
C10:0		0.790		
C10:1	−0.352	0.734		
C11:0		0.838		
C12:0	−0.481	0.751		
C13:0	−0.342	0.720		
C19:0		−0.485		
C20:0		−0.458		0.312
C15:0	−0.353		0.710	
C17:0		−0.494	0.686	
C22:0			0.660	
C14iso		−0.395	0.744	
C15iso			0.860	
C15aiso			0.832	
C16iso		−0.374	0.579	
C17iso	0.345		0.746	
C17aiso			0.679	
Alpha C18:3c9c12c15 (ALA) (n3)			0.472	−0.309
C20:5 (n3)			0.596	
C22:5 (n3)			0.663	
C18:1c12	0.447		−0.618	0.470
C18:2c9c12			−0.461	0.327
C20:3c8c11c14 (n6)				0.697
C20:4c5c8c11c14 (n6)				0.719
C16:1t9	0.394			−0.704
C18:2t9t12	0.318			−0.442
Gamma C18:3c6c9c12 (n6)				−0.577
C21:0		−0.427		−0.444
Variance explained (%)	22.6	16.0	14.9	9.1

**Table 5 animals-09-00111-t005:** Pearson correlation and the two-tailed level of significance of the sum fatty acids (FA) with the expressed factor scores of the four components of Table 4.

Sum Fatty Acids	Comp. 1	Comp. 2	Comp. 3	Comp. 4
Saturated FA	−0.846 (***)	0.360 (***)	0.029 (NS)	0.140 (NS)
Monounsaturated FA	0.816 (***)	−0.413 (***)	−0.097 (NS)	−0.081 (NS)
Polyunsaturated FA	0.664 (***)	−0.173 (*)	0.179 (*)	−0.464 (***)
Short-chain FA	0.403 (***)	0.534 (***)	0.146 (NS)	0.381 (***)
Medium-chain FA	−0.629 (***)	0.651 (***)	−0.050 (NS)	0.030 (NS)
C18:1t	0.890 (***)	−0.023 (NS)	0.102 (NS)	−0.273 (***)
CLA	0.714 (***)	−0.013 (NS)	0.295 (***)	−0.488 (***)
n3	0.362 (***)	−0.118 (NS)	0.428 (***)	−0.255 (**)
n6	0.396 (***)	−0.239 (**)	−0.571 (***)	0.046 (NS)
n6/n3	0.062 (NS)	−0.109 (NS)	−0.741 (***)	0.185 (*)

NS = not significant, * = *p* < 0.05, ** = 0.001 < *p* < 0.05, *** = *p* < 0.001.

**Table 6 animals-09-00111-t006:** Correlation of the background data (season, region, climate, geology, and ecology) with the four components of Table 4. Season and region of production based on ANOVA of the expressed factor scores, and others based on Pearson correlation; the level of significance in between brackets.

Geological/Ecological Data	Comp. 1	Comp. 2	Comp. 3	Comp. 4
Summer	0.677 (***)	0.099 (NS)	0.108 (NS)	−0.085 (NS)
Winter	−0.846 (***)	−0.124 (NS)	−0.135 (NS)	0.106 (NS)
Atlantic	0.103 (*)	0.424 (**)	0.255 (*)	−0.076 (***)
Central	−0.269 (*)	0.054 (**)	−0.250 (*)	0.527 (***)
Pre-Alpine	0.186 (*)	−0.320 (**)	0.072 (*)	−0.445 (***)
Elevation (meters)	0.227 (**)	−0.241 (**)	−0.130 (NS)	−0.224 (**)
Latitude (°E)	−0.173 (*)	0.286 (***)	0.117 (NS)	0.115 (NS)
Longitude (°N)	−0.122 (NS)	0.019 (NS)	−0.184 (*)	0.075 (NS)
Growing degree days (°C)	−0.071 (NS)	−0.083 (NS)	0.346 (***)	−0.073 (NS)
Length of growing season (day)	0.081 (NS)	−0.014 (NS)	0.275 (***)	−0.150 (NS)
Precipitation (mm)	0.242 (**)	−0.231 (**)	0.027 (NS)	−0.404 (***)
Temperature (°C)	−0.127 (NS)	0.074 (NS)	0.328 (***)	0.114 (NS)
Sunshine (hour)	0.092 (NS)	−0.222 (**)	−0.192 (*)	−0.171 (*)

NS = not significant, * = *p* < 0.05, ** = 0.001 < *p* < 0.05, *** = *p* < 0.001.

**Table 7 animals-09-00111-t007:** Significantly different single fatty acids in the summer shop milk of biodynamic and conventional creameries in several European regions based on a paired *T*-test, and the ratio of biodynamic to conventional (B−C)/C × 100 (%).

Fatty Acid	Mean and SD		
Label	Biodynamic	Conventional	*p*-Value	Ratio
Samples (N)	10	10		B to C
C7:0	0.037 (0.008)	0.043 (0.007)	0.043	−13.2
C11:0	0.046 (0.010)	0.059 (0.011)	0.016	−22.6
C15iso	0.295 (0.038)	0.252 (0.021)	0.001	17.4
C16:1t9	0.314 (0.079)	0.265 (0.041)	0.039	18.3
C16:1c9	1.469 (0.141)	1.575 (0.098)	0.021	−6.7
C17iso	0.369 (0.030)	0.335 (0.031)	0.015	10.0
C17:0	0.608 (0.066)	0.537 (0.041)	0.005	13.1
C18:1t11	1.823 (0.819)	1.228 (0.395)	0.021	48.4
C18:1c12	0.113 (0.022)	0.174 (0.047)	0.004	−35.2
C18:1c15	0.092 (0.044)	0.052 (0.031)	0.030	76.4
Alpha C18:3c9c12c15 (ALA) (n3)	0.927 (0.261)	0.602 (0.139)	0.004	54.1
CLAc9t11	1.167 (0.575)	0.806 (0.230)	0.039	44.8
CLAt11c13	0.071 (0.042)	0.042 (0.019)	0.033	66.6
C20:5 (n3)	0.085 (0.013)	0.058 (0.016)	<0.001	47.2
C22:5 (n3)	0.087 (0.010)	0.060 (0.011)	<0.001	44.9
n6/n3	1.37 (0.17)	1.89 (0.37)	0.001	−27.7
Elevation (m)	286 (279)	285 (160)	0.983

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
