# Peer review of "Patterns of Biodynamic Milk Fatty Acid Composition Explained by A Climate-Geographical Approach"

_animals, 2019, doi:10.3390/ani9030111_

Round 1
Reviewer 1 Report
Interesting and important paper. It needs a hard edit. Many sentences are hard to follow, too long. Commas will help in at least 2 dozen places (e.g. lines 203, 283, 285, 297, 325). I will note some places in specific comments below, but by no means all the places the paper's readability will be improved by a solid edit.
The vector analysis is fascinating. I am not qualified to offer judgements re its significance/statistical underpinnings.
Line 27 and 45: Instead of "along the same line.." say instead "are consistent with.."
43: Suggest add overall n6/n3 ratio for biodynamic and conv milk in abstract.
Lines 30 and 47: you say >80% in one place and >85% in another. Confusing. Fix.
56: Suggest "This limits the farmers' ability to..."
71: Close ) missing.
80: "favorable" not "favorite"
83: Suggest "...with n6 FA and contain suboptimal levels of n3 PUFA..."
84: need to add addition cite to Benbrook et al 2013, Organic Production Enhances Milk Nutritional Quality by Shifting Fatty Acid Composition: A U.S.-Wide, 18-month Study," PLOSOne, Vol 8(12): e82429, since the 5.8 n6/n3 ratio comes from that paper
291: delete first "the"
314-315: awkward sentence, rewrite
330: Very interesting observation/paper. Worth saying a bit more about the findings of reference [16]
379-380: suggest revise to read: "...of concentrates per cow. In addition, low-input farms...were more dependent on roughage [15]."
Author Response
Thanks for the review process. In the attached two files, I have answered the points raised by reviewer 1 and 2.

Reviewer 2 Report
The manuscript animals-456710 is a study designed to analyze the differences in milk FA composition of biodynamic farms in Central and Northern Europe and to correlate the FA patterns with the geological and ecological background of the farm location. It is very well written and I really appreciate to review such type of papers: very well described, justified and coherent from the beginning until the end. The only concern that I have and that will be my recommendation to improve data presentation is to use some statistical tools such as Principal Component Analysis or Orthogonal Projections to Latent Structures Discriminant Analysis (OPLS-DA) in order to facilitate the understanding on how FA were separated from winter to summer and from all geographical regions. Tables are OK but a more graphic tool is highly recommended to improve the reader´s understanding. OPLS-DA model can indicate which are the driving forces among the variables and you can then make score plots to visualize the differences if they exist. You can use the loading plot to indicate the variables that express this difference. It is therefore a minor revision needed for this manuscript before its consideration for publication.
Author Response
Thanks for the review process. In the attached file I have given my comments on the questions raised by reviewer 2.

Round 2
Reviewer 1 Report
Ready to go.
Reviewer 2 Report
authors have made changes accordingly